# Nutraceuticals for the Control of Dyslipidaemias in Clinical Practice

**DOI:** 10.3390/nu13092957

**Published:** 2021-08-25

**Authors:** Peter E. Penson, Maciej Banach

**Affiliations:** 1School of Pharmacy & Biomolecular Sciences, Liverpool John Moores University, Byrom Street, Liverpool L3 3AF, UK; P.Penson@ljmu.ac.uk; 2Liverpool Centre for Cardiovascular Science, William Henry Duncan Building, 6 West Derby Street, Liverpool L7 8TX, UK; 3Department of Preventive Cardiology and Lipidology, Medical University of Lodz (MUL), Rzgowska 281/289, 93-338 Lodz, Poland; 4Cardiovascular Research Centre, University of Zielona Gora, 65-046 Zielona Gora, Poland

**Keywords:** nutraceuticals, dyslipidaemia, atherosclerosis

## Abstract

Dyslipidaemias result in the deposition of cholesterol and lipids in the walls of blood vessels, chronic inflammation and the formation of atherosclerotic plaques, which impede blood flow and (when they rupture) result in acute ischaemic episodes. Whilst recent years have seen enormous success in the reduction of cardiovascular risk using conventional pharmaceuticals, there is increasing interest amongst patients and practitioners in the use of nutraceuticals to combat dyslipidaemias and inflammation in cardiovascular disease. Nutraceutical is a portmanteau term: ‘ceutical’ indicate pharmaceutical-grade preparations, and ‘nutra’ indicates that the products contain nutrients from food. Until relatively recently, little high-quality evidence relating to the safety and efficacy of nutraceuticals has been available to prescribers and policymakers. However, as a result of recent randomised-controlled trials, cohort studies and meta-analyses, this situation is changing, and nutraceuticals are now recommended in several mainstream guidelines relating to dyslipidaemias and atherosclerosis. This article will summarise recent clinical-practice guidance relating to the use of nutraceuticals in this context and the evidence which underlies them. Particular attention is given to position papers and recommendations from the International Lipid Expert Panel (ILEP), which has produced several practical and helpful recommendations in this field.

## 1. Introduction

Cardiovascular diseases (CVD) are responsible for an estimated 17.9 million deaths each year and represent the largest overall cause of mortality worldwide [1,2]. Although CVD are a heterogenous range of pathologies, many share a common atherosclerotic pathophysiology. More than four-fifths of these deaths occur as a result of myocardial infarction and stroke [1,2]. Worryingly, one-third of these deaths occur prematurely in people under 70 years of age [1]. However, the picture is not entirely bleak. It has been estimated that over the thirty years to 2016, life expectancy increased by eight years, and the majority of this increase (at least six years) was attributable to cardiology [3]. However, despite the great success of conventional approaches to therapeutics, an enormous disease burden remains. Moreover, in many highly developed countries we may observe a plateau, or even a slight reduction in life expectancy in recent years, which may only be partly attributable to the coronavirus pandemic [4,5].

Nutraceuticals have a role to play in managing and reducing this disease burden [6,7,8]. Until recently, high-quality evidence relating to the safety and efficacy of nutraceuticals has not been available to prescribers and policymakers. However, as a result of recent randomised-controlled trials (RCTs), this situation is changing, and nutraceuticals are now recommended in several mainstream guidelines relating to atherosclerosis and preventive cardiology. This article will summarise recent clinical-practice guidance relating to the use of nutraceuticals in this context and the evidence which underlies them.

## 2. Dyslipidaemias, Inflammation, and Atherosclerosis

The associations between plasma lipoproteins, atherosclerosis and CVD have been topics of intense research since the observations by Gofman [9] and the results of the Framingham research group [10,11] identified associations between low-density lipoprotein cholesterol (LDL-C) and atherosclerotic events. Interventions to reduce LDL-C, including statins [12,13,14,15], ezetimibe [16] and inhibitors of proprotein convertase subtilisin-kexin type 9 [17] have demonstrated remarkable reductions in CV risk when used alone or in combination [18]. Increasing evidence is emerging for newer therapies, including inclisiran [19] and bempedoic acid [20], which can be used to further optimise LDL-C reduction and to enable the achievement of clinical goals, which are still achieved only in about 1/3 patients [21]. 

Furthermore, inflammation, which occurs when dyslipidaemia results in the deposition of LDL-C in the blood vessel wall is increasingly recognised as a therapeutic target [22,23] in the management of atherosclerosis. Additionally, recent research has broadened beyond LDL-C with a renewed interest in the cardiovascular risk conveyed by hypertriglyceridaemia [24,25,26] and hyper-lipoproteinaemia(a) [27]. A wide range of nutraceuticals have been shown to exert biological actions at targets relevant to these processes, and therefore have the potential to treat dyslipidaemias and ameliorate the severity of their consequences. 

## 3. The Use of Nutraceuticals in Dyslipidaemias and Atherosclerosis

The term ‘nutraceutical’ was coined by DeFelice in 1989 as a portmanteau of “NUTRient” and “pharmACEUTICAL”. DeFelice defined nutraceutical as “food, or parts of a food, that provide medical or health benefits, including the prevention and treatment of disease” [1,2] but which are *“produced from foods but sold in pills, powders, (potions) and other medicinal forms not generally associated with food”* [3]. Importantly, nutraceuticals are different from ‘functional foods’ which are *“similar in appearance to conventional foods … consumed as part of a usual diet”* [3]. Nutraceuticals should also be distinguished from other molecules, which come from nature but are not derived from foods. Synthetic molecules, which closely resemble natural molecules also cannot be considered as nutraceuticals. For example, aspirin (derived from willow bark and used in preventative cardiology for its antiplatelet action [28]) and colchicine (derived from the crocus, and which has seen a remarkable renaissance since it has been demonstrated to be effective at addressing inflammation in atherosclerosis [29]) cannot be considered nutraceuticals. Furthermore, although such definitions are somewhat arbitrary, nutraceuticals are generally micronutrients, thus despite extensive research about the health benefits of modifying the proportion of energy derived from different dietary sources [30,31], such macronutrients cannot be considered as nutraceuticals. 

The development of nutraceuticals, therefore, attempts to use biologically active drug-like molecules from nature to prevent or treat disease. However, instead of consuming such molecules in the diet (by eating foods known to contain them), nutraceuticals enable control of dose, quality, and composition of formulations, thus enabling the principles of Good Manufacturing Practice (GMP) to be applied, using the same approach as is used in the production of conventional pharmaceutical products.

The International Lipid Expert Panel (ILEP) position paper on lipid-lowering nutraceuticals in clinical practice for the first time provides a comprehensive and detailed overview of a very wide range of nutraceuticals with the potential to elicit favourable effects on plasma lipoproteins. These include inhibitors of intestinal cholesterol absorption (plant sterols and stanols, soluble fibres, chitosan and probiotics); inhibitors of liver cholesterol synthesis (red yeast rice, garlic, pantethine, bergamot and policosanols); compounds that promote LDL-excretion (berberine, green tea extracts, soy and lupin); and a range of other nutraceuticals including ω-3 fatty acids, spirulina and curcumin [32]. Helpfully, the position paper makes clear and detailed recommendations for each agent and classifies the recommendations according to the strength of recommendation and level of evidence supporting it. The experts from the panel that was founded in 2015 [33], aimed in this paper to present only these nutraceuticals that have real beneficial effect on LDL-C based on high quality data; they also discussed in detail their safety (nutrivigilance), and present recommendations that patients might benefit from the most and in what clinical settings [32]. 

A more recent position paper by the same group takes a very similar approach but focuses instead on the evidence supporting the anti-inflammatory actions of nutraceuticals [34], a topic of great importance during the current pandemic and in a situation where there is a lack of effective drugs to reduce inflammatory markers [29,35]. Taken together, these two documents provide a wealth of useful information relating to the management of plasma lipids and the associated risk of atherosclerotic disease in clinical practice. 

It is beyond the scope of the present article to review this evidence again; however, the reader is directed to the ILEP papers (including a supplementary paper on the role of nutraceuticals in heart failure patients [36]). The evidence supporting the clinical use of three well-evaluated nutraceuticals (red-yeast rice, ω-3 fatty acids and phytosterols) is described in the sections below, and brief mechanistic details for these agents are provided in Figure 1. 

### 3.1. Red Yeast Rice

Red yeast rice (RYR) is a traditional fermented ingredient of Chinese food which contains a bioactive molecule called monacolin K, a naturally occurring inhibitor of 3-hydroxy-3-methyl-glutaryl-coenzyme A reductase, the target of statins. As such, the consumption of red yeast rice and its nutraceutical preparations reduces plasma LDL-C [37]. Remarkably, an RCT that evaluated a nutraceutical extract of red yeast rice in 5000 individuals in China reported a 45% relative reduction in cardiovascular events [38]. If this effect size can be achieved in other populations, it suggests that RYR preparations might have an important contribution to make to CVD risk reduction. 

In the past, concerns have been raised about the safety of red-yeast rice preparations. These generally relate to the batch-to-batch variability in monacolin content and the presence of toxic by-products of rice fermentation (citrinin) [39]. These issues can be resolved through the employment of the principles of GMP. Indeed, a recent nutrivigilance study of a high-quality nutraceutical preparation containing red-yeast rice and other lipid-lowering nutraceuticals found that only 0.037% of 2,287,449 consumers reported adverse events [40]. The recent draft recommendations to European Food Safety Authority (EFSA) suggest that RYR preparations should contain no more than 3 mg of monacolin K. 

### 3.2. ω-3 Fatty Acids

Polyunsaturated ω-3 fatty acids (PUFAs), which are derived from oily fish, have been the subject of intense recent research. Eicosapentaenoic acid (EPA) and docosahexaenoic acid (DHA) reduce circulating concentrations of triglycerides [41] and therefore have been evaluated for the potential to reduce the CV risk associated with hypertriglyceridemia. Unusually for nutraceuticals, polyunsaturated ω-3 fatty acids have been evaluated in numerous cardiovascular outcomes trials (CVOTs), however, the results of the trials have not been consistent, resulting in considerable discussion within the scientific and medical community. 

The VITAL study [42] and ASCEND trial [43] randomised patients to receive placebo or a preparation of ω-3 fatty acids approved by the American Heart Association (AHA) (460 mg EPA, 380 mg DHA). The trials were similar, although a higher risk population was recruited in ASCEND. Neither trial showed a difference in composite CVOTs over a relatively long period of follow-up (7.4 years in ASCEND, or 5.3 years in VITAL). Recently, STRENGTH, a similar trial (however, with the high dose of mixed omega-3 fatty acids of 4 g/day), was halted based on an interim analysis, which suggested no likely benefit of the intervention a combination of EPA/DHA) [26,44].

Conversely, a high dose (4 g) preparation of pure EPA (prepared as an ethyl ester and known as ‘icosapent ethyl’) has demonstrated benefit on physiological parameters and cardiovascular outcomes [24,25]. The EVAPORATE trial demonstrated [45] that icosapent ethyl resulted in atherosclerotic plaque regression (measured using coronary computed tomography angiography). However, more importantly, the REDUCE-IT randomised, placebo-controlled trial evaluated icosapent ethyl in 8179 participants and demonstrated a 25% relative reduction in cardiovascular events over five years [24,25]. However, the most recent meta-analysis suggest (similarly to the one presented by authors [46]), that omega-3 FAs reduced CV mortality and improved CVOTs, however, with more prominent cardiovascular risk reduction with EPA monotherapy than with EPA+DHA [47].

There are some safety concerns for omega-3 acids in the available studies. In the REDUCE-IT trial, an increase in the rate of hospitalization for atrial fibrillation and peripheral edema in the icosapent ethyl vs. placebo group were observed [24,25]. 

### 3.3. Phytosterols

Phytosterols and phytostanols have been widely used in nutraceuticals and in functional foods (such as phytosterol-enriched yoghurts and spreads). Unfortunately, there is no randomised-controlled trial data evaluating the effects of these compounds on hard clinical outcomes [48]. As such, the evidence basis for these interventions relies on surrogate outcomes. Nevertheless, such evidence is convincing. A meta-analysis of 113 RCTs demonstrated a dose-dependent effect of plant sterols on LDL (by even 15%), with optimal daily doses above 2 g [49]. 

Also, in case of phytosterols some safety concerns need to be mentioned. They can lead to hypercholesterolemia and elevated CVD risk in some rare genetic abnormalities of the ABCG5/ABCG8 transporters [50].

## 4. General Recommendations for Nutraceuticals in International Guidelines

International guidelines vary in the extent to which they recommend nutraceuticals. The 2019 European Society of Cardiology/European Atherosclerosis Society (ESC/EAS) Guidelines for the management of dyslipidaemia give Class A recommendations to the three classes of nutraceuticals described above (red-yeast rice, ω-3 fatty acids and phytosterols) [48]. The American College of Cardiology/American Heart Association (ACC/AHA) guidelines for the primary prevention of cardiovascular disease [51] and the management of blood cholesterol [52] do not recommend any nutraceutical preparations. In the United Kingdom, the National Institute for Health and Care Excellence (NICE) guidelines do not make any recommendations favouring nutraceuticals [53] and specifically advise against the use of ω-3 fatty acid compounds [53]; however, it should be noted that these guidelines precede the publication of IMPROVE-IT and EVAPORATE, and therefore do not reflect the latest evidence relating to EPA. The guidelines are currently undergoing a process of update and revision.

More specifically, ILEP recommendations, which are focused directly on this issue, have suggested a range of specific situations, in which nutraceuticals may be considered in the management of dyslipidaemias and atherosclerosis. These are discussed briefly below (Section 5, Table 1), and the reader is encouraged to access the relevant ILEP position papers for practical advice on the control of dyslipidaemias in clinical practice. 

## 5. Specific Situations in Which Nutraceuticals May Be Considered

### 5.1. Statin Intolerance

Statin intolerance occurs when adverse effects on statin therapy limit the ability of an individual to take statins at guideline-recommended doses [56,57]. Many adverse symptoms reported by patients receiving statin therapy are coincidental and not caused by statins. Therefore, careful diagnostic workup must be undertaken to exclude other causes of symptoms [58,59] (see also Section 5.2 ‘Nocebo/Drucebo effect’, below). However, in the rare cases of complete statin intolerance (usually up to 5% where patients cannot tolerate any statin at any dose) or the more common situation of partial intolerance (where a patient can tolerate a statin, but not at a sufficiently high dose to reach their treatment targets), additional lipid-lowering therapies are necessary. Ideally, such patients may receive PCSK9 inhibitors; however, these are not available or reimbursable in all jurisdictions, and therefore, with other drugs still unavailable and not reimbursed (bempedoic acid and inclisiran), nutraceuticals may be considered in combination with other lipid-lowering drugs. Helpful guidelines, including classification of level and class of evidence, have been published by ILEP relating to the use of nutraceuticals in this context [50].

Nutraceuticals may also be of benefit in patients who are nonadherent to statin therapy (as statin intolerance is the most common reason of statin nonadherence), as well as those who are not willing to use statins, despite indications and the physician’s attempts to convince them (statin deniers). Unfortunately, this group (often young people who feel healthy in general) can constitute even 5–7% of patients who should be on statin therapy [54,58,60].

### 5.2. Nocebo/Drucebo Effect

In relation to statin intolerance (see Section 5.1 ‘Statin intolerance’ above), it is important to note that most reported muscle pain in statin therapy is attributable to the nocebo/drucebo effect and occurs as a result of the patient’s expectation of adverse effects [61]. Whilst the ultimate aim of treatment should be to help patients to use life-saving statins [62], nutraceuticals may be useful in the control of plasma lipids in these patients. Forthcoming ILEP recommendations will address this issue in detail [54].

### 5.3. Patients Considered ‘Low-Risk’ by Conventional Risk Scores

In most jurisdictions, patient eligibility for statin therapy is decided based on their risk of cardiovascular disease, usually calculated over a 10-year period, based on their demographic characteristics and physical and biomarker measurements. Such an approach almost certainly underestimates the life-long risk of cardiovascular disease in younger individuals with dyslipidaemias but without other risk factors. Epidemiological and interventional trials have unambiguously demonstrated that CVD risk is a function of lifelong exposure to LDL-C, and this can be helpfully summarised as ‘lower is better for longer’. ILEP recommendations have been produced to enable the optimal management of patients in this situation. Initially, LDL-C should be controlled by lifestyle interventions (including exercise, bodyweight reduction, smoking cessation and adherence to a diet low in saturated fat and high in plant protein and fibre); however, if adherence to this regimen is poor, or LDL-C is not sufficiently reduced, then nutraceuticals (especially in the form of polypills) may be considered [55].

### 5.4. Optimisation of Therapy in High-Risk Patients

ILEP has also produced advice about the use of nutraceuticals in patients at high risk of CVD. They summarise the potential uses of nutraceuticals as follows: (1) managing residual risk associated with lipids other than low-density lipoprotein cholesterol (for example, risk mediated by triglycerides may be ameliorated by EPA (see Section 3.2 ‘ω-3 fatty acids’)); (2) managing non-lipid-mediated residual risk (such as inflammatory risk [34]); (3) optimising LDL-C treatment in statin intolerance (see Section 5.1 ‘Statin intolerance’ and Section 5.2 Nocebo/Drucebo effect, above); (4) optimising LDL-C treatment when add-on therapies for statins are not available (e.g., with the limited reimbursement criteria); (5) as adjuncts to lifestyle for individuals at high lifetime risk of atherosclerotic cardiovascular disease (ASCVD) [7].

### 5.5. Patient-Initiated Nutraceutical Use

A wide variety of nutraceutical preparations are available for patients to purchase over the counter. Prescribers and physicians should always question patients about nutraceutical use as part of routine history taking. This is important to avoid potential adverse effects relating to nutraceutical use. Whilst most preparations are thought to be generally safe, long-term safety data is often not available (see information above). Harm may also result as a result of batch-to-batch variation in the content of the active ingredient in poor quality preparations or because of contamination of preparations with unwanted ingredients [39]. Both problems can be overcome by adherence to the principles of GMP; however, the relatively light regulatory framework for such products in some jurisdictions means that low-quality preparations may reach the market. When patients do choose to use nutraceuticals, it is important that this occurs as a supplement to evidence-based guideline-directed therapies and not as a replacement for them. 

## 6. Conclusions

The development of nutraceuticals from the micronutrient components of food presents an opportunity to target dyslipidaemias and atherosclerosis through direct effects on plasma lipids, and through the modification of pathophysiological processes elicited through atherogenic lipoproteins. Until recently, limited evidence has been available to evaluate the efficacy of nutraceuticals on clinical outcomes and their safety at doses above those usually consumed in the diet. However, recent data from RCTs and nutrivigilance studies has changed this picture. The strongest evidence exists for plant sterols and stanols, eicosapentaenoic acid and red-yeast rice. As such, guidelines and position papers (especially those produced by the ILEP) have recommended roles for nutraceuticals. Whilst many practitioners prescribe or recommend nutraceuticals in their practice, the increasing availability of such preparations and their use by patients makes it imperative that all practitioners treating dyslipidaemias and atherosclerosis are aware of major nutraceuticals, their indications, and the evidence supporting their use. It should be stressed, however, that nutraceuticals should only be used to supplement, not to replace guideline-directed evidence-based therapeutics.

## Figures and Tables

**Figure 1 nutrients-13-02957-f001:**
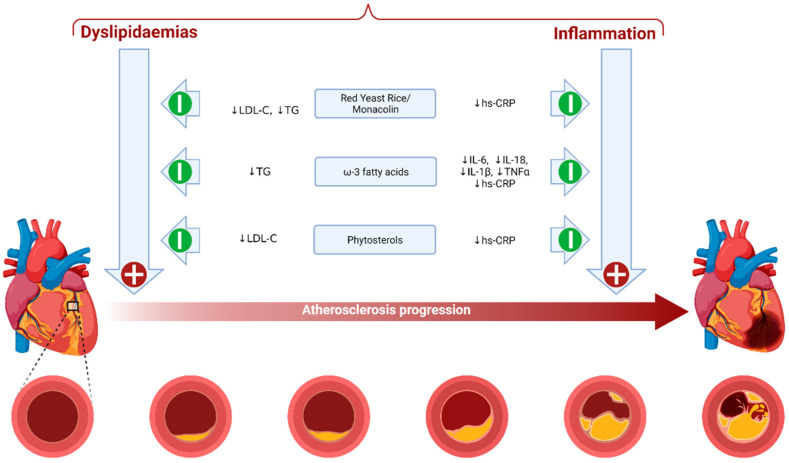
Summary of biological mechanisms involved in the anti-atherosclerotic actions of key evidence-based nutraceuticals. Image Adapted from “Atherosclerosis Progression”, by BioRender.com (2021). Retrieved from https://app.biorender.com/biorender-templates (Accessed date: 23 July 2021).

**Table 1 nutrients-13-02957-t001:** Summary of recommendations on the use of nutraceuticals in dyslipidaemias and atherosclerosis.

Organisation	Patient Population	Recommendation	Ref
ILEP	Statin intolerance	Nutraceuticals may be used in combination with other lipid-lowering drugs.	[50]
ILEP	Nocebo/drucebo	Nutraceuticals may be used in combination with other lipid-lowering drugs.	[54]
ILEP	Low CVD risk	Nutraceuticals may be appropriate to control lipids in patients ineligible for statins (and for those not willing using statins).	[55]
ILEP	High CVD risk	Nutraceuticals may be used in combination with other lipid-lowering drugs and may be useful to control residual risk.	[7]

Abbreviations, CVD, cardiovascular disease; ILEP, International Lipid Expert Panel.

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
