# Peer review of "Nutraceuticals for the Control of Dyslipidaemias in Clinical Practice"

_nutrients, 2021, doi:10.3390/nu13092957_

Round 1
Reviewer 1 Report
The manuscript by Penson and Banach is an effective high-level literature review (position paper and recommendations from the International Lipid Expert Panel) on the use of nutraceuticals in clinical practice.
This work is important in light of the fact that in the International Guidelines (ESC / EAS 2019) little space is reseved to dietary and nutraceutical therapies, while in patients with low cardiovascular risk and intolerant / refusing statins, they can represent the cornerstone of lipid-lowering therapy. The revision is well done.
Perhaps something more extended could be added about berberine which has shown lipid-lowering nutraceutical efficacy documented in some studies, even if more limited.
Author Response
Response to Reviewer:
Thank you for your positive feedback on our paper. Thank you for your suggestion on berberine. Its role has been already strongly emphasized in all cited ILEP recommendations. Here in this paper we focused on those three with the largest EBM (including CVOT data), as well as those already mentioned in European recommendations. Adding more data on berberine would mean we should do the same for the other well-recognized like policosanol, bergamot, curcumin, sterols, etc. Therefore, we would like to stay with the current version of the paper, if it is acceptable.
Reviewer 2 Report
The review “Nutraceuticals for the control of dyslipidemias in clinical practice” summarizes recent clinical-practice guidance related to the use of nutraceuticals for the control of hyperlipidemia. Particular attention is given to position papers and recommendations from the International Lipid Expert Panel (ILEP), which has produced several practical and helpful recommendations in this field.
The article is appropriately organized and well written.
I suggest the correction of minor mispellings as
Line 72: The term “nutraceutical” was…
Line 139: 3.2.ω3 fatty acids
Line 264-265: please rephrase the sentence
Author Response
Thank you for all your valuable comments. We have revised the text accordingly.